# Deep Learning Model for Selecting Suitable Requirements Elicitation Techniques

**Hatim Dafaalla** [1,*], **Mohammed Abaker** [1], **Abdelzahir Abdelmaboud** [2], **Mohammed Alghobiri** [3], **Ahmed Abdelmotlab** [3], **Nazir Ahmad** [1], **Hala Eldaw** [4] **and Aiman Hasabelrsoul** [5]

[1] Department of Computer Science, Applied College, King Khalid University, Muhayil 61913, Saudi Arabia
[2] Department of Information System, College of Science and Art, King Khalid University, Muhayel 61913, Saudi Arabia
[3] Department of Management Information System, College of Business, King Khalid University, Abha 62529, Saudi Arabia
[4] Department of Information System, College of Computer Science & Information Systems, Al Jouf University, Sakaka 72388, Saudi Arabia
[5] Department of Business Administration, Applied College, King Khalid University, Muhayel 61913, Saudi Arabia
* Correspondence: hibrahem@kku.edu.sa

**Abstract:** Requirement elicitation represents one of the most vital phases in information system (IS) and software development projects. Selecting suitable elicitation techniques is critical for eliciting the correct specification in various projects. Recent studies have revealed that improper novice practices in this phase have increased the failure rate in both IS and software development projects. Previous research has primarily relied on creating procedural systems based on contextual studies of elicitation properties. In contrast, this paper introduces a deep learning model for selecting suitable requirement elicitation. An experiment was conducted wherein a collected dataset of 1684 technique selection attributes were investigate with respect to 14 elicitation techniques. The study adopted seven criteria to evaluate predictive model performance using confusion matrix accuracy, precision, recall, F1 Score, and area under the ROC curve (AUC) and loss curve. The model scored prediction accuracy of 82%, precision score of 0.83, recall score of 0.83, F1 score of 0.82, cross-validation score of 0.82 ($\pm$ 0.10), One-vs-One ROC AUC score of 0.74, and One-vs-Rest ROC AUC score of 0.75 for each label. Our results indicate the model's high prediction ability. The model provides a robust decision-making process for delivering correct elicitation techniques and lowering the risk of project failure. The implications of this study can be used to promote the automatization of the elicitation technique selection process, thereby enhancing current required elicitation industry practices.

**Keywords:** requirement elicitation; elicitation technique selection; deep learning; neural network

## 1. Introduction

The requirement elicitation process represents the first phase of every software and IS project development. The importance of this phase has been widely discussed through countless research and survey reports. Its susceptibility impact can be viewed in the report by the Standish group [1], depicting a 31% failure ratio for software development projects. Moreover, 51% of these undergo serious challenges that extend the duration of the project, further increasing budget costs. Successful software delivery demands correct software specifications and requirements through applying appropriate elicitation techniques. Therefore, selecting a proper elicitation technique requires extracting more accurate and complete requirements that reflect stakeholders' actual desires. Hence, each technique has its strengths and weaknesses depending on the case scenario of the requirement. According to the authors of [2], most software engineers select an elicitation technique for several reasons, whereas they typically select one specific strategy (i.e., favoring one method for all

possible scenarios). Otherwise, they tend to presume the technique's effectiveness during the application process. The literature for the requirement elicitation phase is rich with numerous contributions that vary from primitive to advanced applications of technology and methodologies. However, studies show that most of the research published in this field is focused on the level of requirement identification and classification [3]. Unquestionably, successful implementation of the elicitation technique selection process using machine learning is tangible in the requirement elicitation field. Nevertheless, studies have uncovered the limitations practical machine learning applications in the field [4–6]. We need to enhance the requirement elicitation process with more than just automated operation using machine learning. Instead, we need to create a model that opts to learn and think like humans. Hence, this paper aims to create a requirement technique selection model using deep learning technology that reduces the software engineers' intervention in the technique selection process with a more robust and effective alternative that can generate more accurate decisions. Eventually this could lead to more precise requirement reports reflecting the actual needs of stakeholders, thus increasing the success ratio of the ongoing software development project. Thus, the main contributions of our model are as follows:

- Automating the technique selection process to reduce human error;
- Building a robust decision-making model;
- Producing proper requirements and increasing the success ratio of IS projects.

The remaining part of the paper is organized as follows. Section 2 presents the related study. Section 3 illustrates the strategy used to implement the proposed methodology in order to develop a deep learning model for elicitation technique selection. Section 4 presents the deep learning model implementation results and model validation. Section 5 discusses the model results. Section 6 presents the conclusions of our research in this paper.

## 2. Related Study

This section focuses on the requirement elicitation phase, and more specifically the elicitation technique selections. Moreover, this section also focuses on approaches adopted by the researchers for the elicitation technique selection used in the industry.

Requirement elicitation is a vital phase in the requirement engineering process. The software and IS project development depends highly on the requirement elicitation practice [7]. Selecting a proper requirement technique is essential for collecting accurate requirements from stakeholders. However, selecting a proper technique is difficult [8] for several reasons, including the diversity of the technique's property, software engineer experience, and the decision-making process for selecting the method. However, any technique's case selections should not be based on preference or trial rather than experience. Thus, research has always attempted to provide a more solid decision-making process that utilizes available technology. Hence, an attempt by N.R. Darwish et al. [9], one of the pioneers of implementing artificial neural network (ANN) for technique selection schemes, was effective to some degree in reducing the human involvement factor (thus reducing human error). However, classification attributes are tailored according to a project's characteristics. Therefore, they do not reflect the properties and applications of the technique involved. However, technology and algorithmic implementation was not the only tool used to develop solutions. in P. Vitharana et al. [10], there are two mental models developed for enhancing the requirement elicitation process. Moreover, this research has emphasized the value of accurate system requirements. However, this presented study has several limitations regarding the validity of the experiment conducted. Moreover, attempts made by researchers such as I. Bodnarchuk et al. [11] discussed the importance of assuring the software quality in software production by improving the decision selection process through the implementation of the analytical hierarchy process (AHP) and goal function. However, this model presented a different approach to requirement elicitation technique selection. Nevertheless, it did not provide a systematic process to automate the selection process using more authentic operations. Nevertheless, the limitations of the work of I. Bodnarchuk et al. were addressed to some degree in H.M.E. Ibrahim et al. [12],

who attempt to automate the requirement elicitation technique selections by implementing machine learning. The model's results were very promising considering the dataset's smaller scale and the particular experiment conducted (which limit the chances of generalizing the results). Furthermore, this model is considered the kernel of the current presented model. Moreover, K. Gupta, and A. Deraman [13] argued one should produce unambiguous requirements reflecting the software desired functionality by selecting the correct elicitation technique that efficiently induce a wider range of requirements. Thus, their software requirement ambiguity avoidance framework (SRAAF) was an attempt to enhance the elicitation process to reduce the ambiguity ratio. However, it was not qualified as an algorithmic solution, lacking in terms of its level of automation. Likewise, F. Hujainah et al. [14] proposed a semi-automated stakeholder quantification and prioritization technique for requirement selection in software system projects (StakeQP). However, their presented model was fixed on stakeholders involved as the benchmark for producing the correct requirements. Moreover, the lack of complete automation has weighed on the time factor. Hence StakeQP was considered a time-consuming process. After that, S. M. Giraldo et al. [15] explored over 280 references, 16 experts in the field of requirement elicitation, and records of 32 companies' projects to gather selection techniques attributes and measure their capabilities and application in the elicitation scenarios. The study data provided valuable insights on technique characteristics and paved the way to transition to more modern applications. One of these modern applications can be found in the work of J. Li et al. [16], who attempted to implement an analytic network process method for selecting the elicitation technique. Furthermore, the study has distinguished the application of the analytical network process in the elicitation technique selection regarding project attributes. Again, H.M.E.I. Dafalla et al. [17] provided another model to classify the elicitation technique based on the attributes through the k-nearest neighbor's algorithm implementation. The results were promising, but this success was attributed to the small size of the implemented dataset. Furthermore, the study validated the results using only the similarity rates. Next, M.B. Rehman, H.M.E.I. Dafallaa. [18] presented requirements conflict resolution and communication model for the telecommunication sector. The model implemented the normalized cross-correlations function (NCCF) to detect requirements conflict. Moreover, the model validated the results using the standard error (SE) function. Although the model was successful, it was based on a statistical assertion. However, the model overlooked the technique selection process. S. Panichella and M. Ruiz further attempted to automate the requirement elicitation process [19] by presenting a requirements-collector tool. Although machine learning and deep learning were used to implement the mechanism, much diligence was given to the text classification phase of the process [20,21] which overlooks the technique selection decision-making process. Alternatively, H. Saeeda et al. [22] presented a framework for improved software requirements elicitation. The proposed framework was implemented in a real-life Norway-based IT project. Although, the result of the framework was statistically helpful, a further transition towards automation using machine learning and deep learning was needed. Next, the negative impact of improper requirement selections was investigated by D. Mougouei et al. [23]. As a result of the study, the authors presented a partial selection of the software requirements model using a fuzzy method. The experiments revealed enhanced requirement reports that contributed to building the desired software. However, the study needs further investigate the negative influence of the applied method. Henceforth, the research community has shifted its focus towards more robust technologies in search of a solution. Thus, M. Naumcheva. [24] developed deep learning model in software requirements engineering. The following study presented a transitional state in the type of technology applied to the field. However, the conducted study was implemented on the classification phase of the requirement only. Similarly, B. Li, Z. Li, and Y. Yang. [25] presented a novel study using a deep neural network to automate the extraction of non-functional requirements. The model is yet another transitional state used in requirement engineering, and demonstrated promising results. However, the proposed model overlooks the elicitation technique selection decision-making process.

Conversely, J.D. Sagrado and I.M.D Águila. [26] thought to assist the requirements process by developing a selection by clustering model. The produced results of the three different experiments were promising. However, the study posed a few threats to the validity related to the experimental methodology. Finally, H. Elhassan et al. [27] proposed a requirement conflict detection model using machine learning that reduces the processing overhead through automation sequence operation. Although the results were promising, the study revealed validation threats concerning the size of the conducted experiment. Moreover, the proposed model has tended to focus on requirement conflict detection rather than the selection of technique decision-making process.

The literature review was rich with various successful approaches and studies about elicitation technique selection. From the literature review timeline, we can notice the gradual development of the field in context with the development of technology. Undoubtedly, the outcome of these experimentations was promising, as we can note from the illustrated analysis in section two. However, drawbacks and limitations exist in terms of the opportunities and goals for this paper. The illustrated approaches in this literature section clearly show the lack of robustness and structural decision-model processes regarding technique selection. Moreover, there is a lack of dataset compatibility in regard to technique attributes. Furthermore, the automatizing process is often employed infrequently. As such, our goal is to design an innovative, robust decision-making model which automate the selection of elicitation techniques using deep learning technology, thus reducing the recurrence of human errors, improving software and IS project practices, and lowering the risk of failure.

## 3. Methodology and Materials

This section presents an overview of the proposed methodology for developing the deep learning model for elicitation technique selection, the data collection process, model implementation, and model analysis and results.

*The Methodology Strategy*

This section describes the methodology strategy used in the deep learning technique selection model. The proposed model consists of three major phases, as shown in Figure 1. Data preparation, as phase one, will start by selecting most influential technique attributes affecting the elicitation performance. Next, the chosen technique selection attributes from various sources are surveyed. Data preprocessing will be conducted to initialize, format, and map the technique attributes' surveyed weight to build the dataset. Lastly, the dataset will be scaled for faster convergence by gradient descent. Model training as phase two will implement the multilayer perceptron (MLP), a feedforward artificial neural network that generates a set of outputs from a set of inputs using the neural network library of SciKit-Learn, thus creating an instance for the model by defining the three hidden layers and assigning the number of neurons at each layer. Finally, model validation phase three will use confusion matrix accuracy, precision, recall, F1 Score, cross-validation, the area under the ROC curve (AUC), and the loss curve to find an optimal model with the best performance.

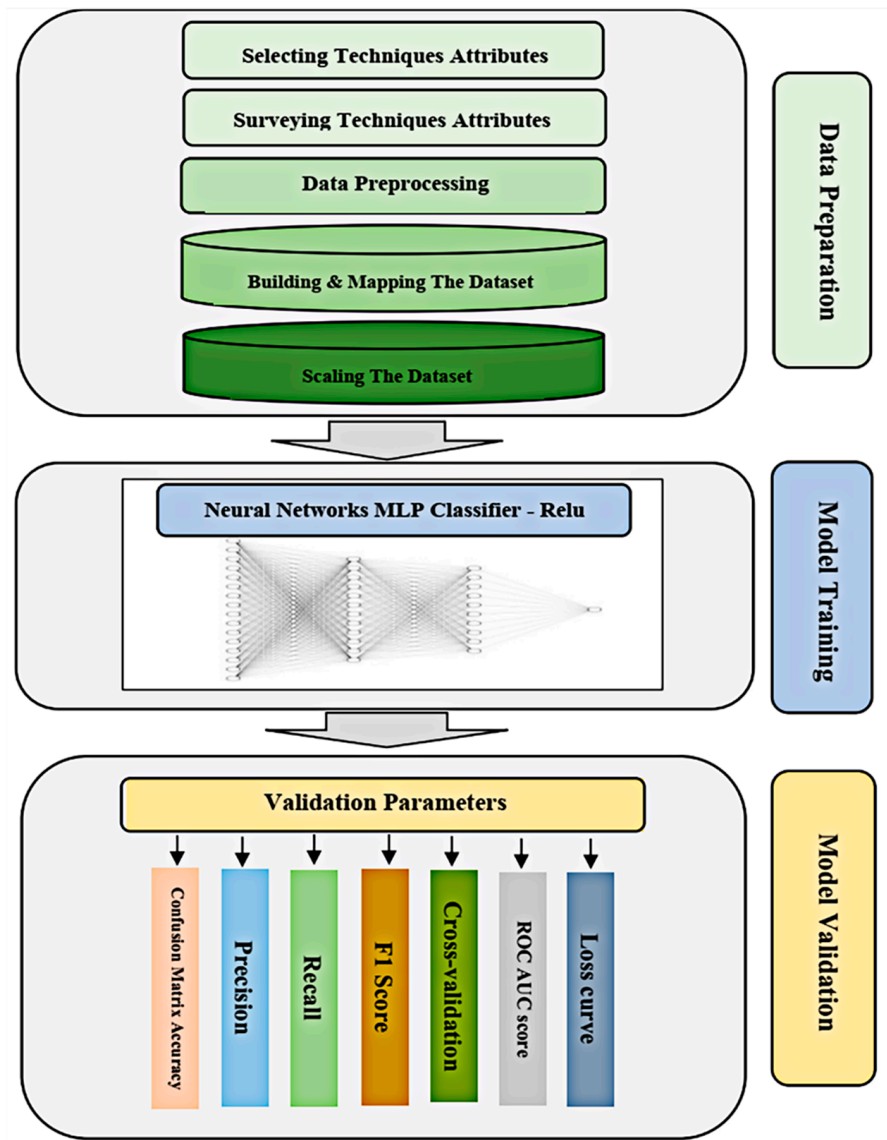

**Figure 1.** Methodological flowchart.

## 4. Data Preparation

### 4.1. Technique Selection Attributes

Technique selection attributes are very significant in this study for their role in the decision-making process. Thus, there is an urgent need to define the most influential technique attributes affecting the elicitation performance to be considered the primary parameter for a proposed dataset. Luckily, the field of requirement engineering is rich with various conducted studies and experimentation on a different set of technique attributes. For example, a significant study by [2] D. Carrizo et al. recorded 34 influential technique attributes. This research design is the backbone of successful model applications such as in [12,13,15,16]. However, this study is based on the technique attributes of our previous published models [12,17]. Thus, this study will use four attributes: analyst experience, technique attribute, technique time, and level of information. The selected attributes are supported by the technique selection models [2,12,13,15,16], as shown in Table 1.

**Table 1.** Technique selection attributes.

| Attributes | Classification |
|---|---|
| Analyst Experience | Classifies the system analyst experience, the level of involvement in software development projects and familiarity of the system analyst with the elicitation technique. |
| Technique Attribute | Classifies the range of individuals that could be accommodated by the elicitation |
| Technique Time | Classifies the time duration of the elicitation technique |
| Level of Information | Classifies the scale of the information extractions |

*4.2. Data Collection*

We have performed a comprehensive data collection process from various data sources to maximize the existing datasets in [12,13,15,16] through survey inquires design to generate a two-dimensional matrix to populate and preprocess the values of the technical parameters for the 14-elicitation techniques (as shown in Table 2). Thus, we successfully built a dataset consisting of 1684 technique selection attributes samples for the 14 elicitation techniques from Saudi Arabian companies. Moreover, the scaling and mapping process was conducted on the dataset to transform the records to the normalized numerical weights expressed in Equation (1), thus enhancing the deep learning neural network mapping process of input variables to an output variable. Finally, the dataset was verified for correctness and duplication.

$$z = \frac{X - \mu}{\sigma} \tag{1}$$

where calculate **a** standardized value (a z-score), Mu ($\mu$) the mean, Sigma ($\sigma$) the standard deviation, and **X** the observation.

**Table 2.** Technique selection parameters.

| Analyst Experience | Technique Attribute | Technique Time | Level of Information |
|---|---|---|---|
| Low | Single | Low | Low |
| Medium | Group | Medium | Medium |
| High | Large Group | High | High |

## 5. Deep Learning Model

The proposed deep learning model for selecting suitable requirements elicitation techniques, illustrated in Figure 2, is a neural network-based configuration for elicitation techniques selection precision and automation. The proposed model consists of 3 hidden layers, each containing 100 neurons. Each line that connects these inputs to the neuron is assigned a weight. This leads to 1 singular output unit of a suitable requirement elicitation technique nomination out of the 14 elicitation techniques.

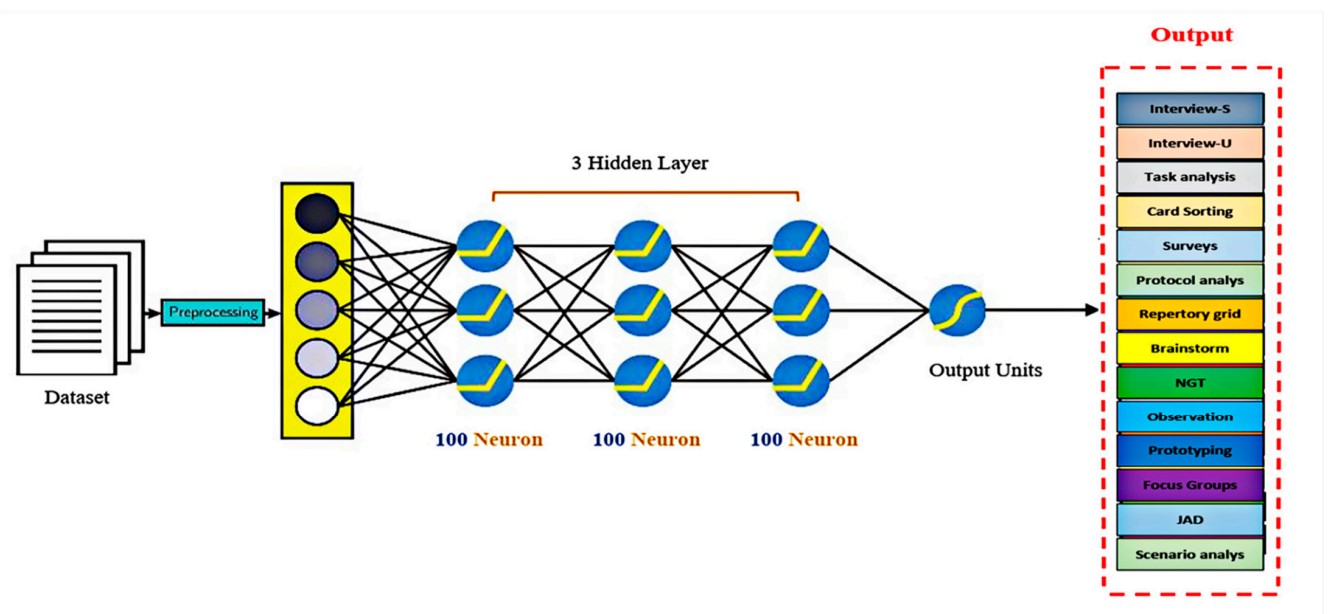

**Figure 2.** Deep learning model design.

### 5.1. Artificial Neural Networks Based Model

Requirement elicitation technique selection is a complex process that has a huge impact on the progress of IS and software development projects. As such, a higher level of experience is required. Therefore, in this paper we will use of artificial neural networks to imitate experts' decision-making processes in the field.

The proposed model, Figure 2, illustrates the deep learning model's architecture. This consists of an input layer with four parameters, three hidden layers, each with a hundred neurons, and an output layer representing the requirement elicitation techniques. Each neuron of the hidden layers and the output neuron possess corresponding biases. Each neuron of the hidden layers receives its output from every neuron of the previous layers and transforms these values with a weighted linear summation expressed as follows in Equation (2).

$$\sum_{i=0}^{n-1} w_i = w_0 x_0 + w_1 x_1 + \ldots + w_{n-1} x_{n-1} \tag{2}$$

where **n** is the number of neurons of the layer and $\mathbf{w_i}$ corresponds to the ith component of the weight vector. The output layer receives its values from the last hidden layer. We needed to employ a more sensitive activation function in this model to avoid saturation. Thus, we implemented a rectified linear unit (ReLU). As such, plaining the network training and allowing the model to account for non-linearities and specific interaction effects, thus improving the performance of the neural network model. The ReLU activation function is expressed as follows in Equation (3), and will return the same positive values in the case of negative inputs.

$$f(x) = \max(0, x) \tag{3}$$

Finally, the optimization algorithm, gradient descent, and backpropagation are expressed as follows in Equation (4), which will be run to minimize the error values between predicted and actual results.

$$X = X - lr * \frac{d}{dx} f(X) \tag{4}$$

where X is the input, f(X) is the output based on X, and lr is the learning rate.

### *5.2. Analysis and Results*

This section analyzes the deep learning model for selecting suitable requirement elicitation techniques. In this model, we intend to analyze the performance of the deep learning model using the confusion matrix. The model started by preprocessing and scaling the dataset using Scikit-learn, the machine learning library for python programming. The dataset comprises 1684 technique selection attribute samples for the 14 elicitation techniques. These samples will be split into two subsets to estimate the model performance. The first subset was used to train the model in 70% (1178) samples of dataset records. The second subset will be used for testing purposes in 30% (506) samples of dataset records to compare the model prediction with the expected.

Figure 3 illustrates the confusion matrix report of the deep learning model for the 506 testing samples. There are cases the model predicted yes for the 14 classified elicitation techniques (true positives (TP)); predicted no (true negatives (TN)); falsely predicted yes (false positives (FP)); and falsely predicted no (false negatives (FN)). These four metrics will allow us to calculate the performance metrics (such as precision to measure the model's ability to return only the data points in a class as follows in Equation (5); recall to measure the model's ability to identify all data points in a relevant class as follows in Equation (6); F1 score to reflect how reliable the model is in classifying samples, as follows in Equation (7); and accuracy to measure the model the ratio of correctly predicted elicitation technique, as follows in Equation (8) and as shown in Table 3).

$$\text{Precision} = \frac{\text{TP}}{\text{TP} + \text{FP}} \tag{5}$$

$$\text{Recall} = \frac{\text{TP}}{\text{TP} + \text{FN}} \tag{6}$$

$$\text{F1 score} = 2 * \frac{\text{Precision} * \text{Recall}}{\text{Precision} + \text{Recall}} \tag{7}$$

$$\text{Accuracy} = \frac{\text{TP} + \text{TN}}{\text{TP} + \text{TN} + \text{FP} + \text{FN}} \tag{8}$$

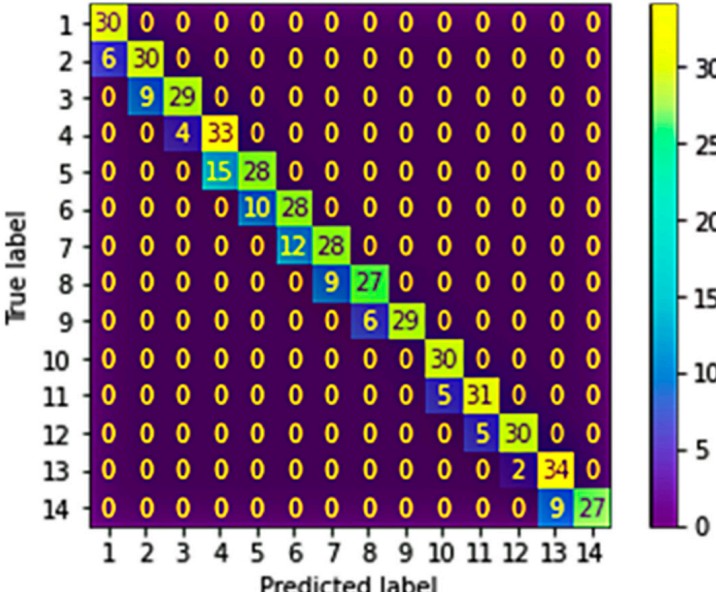

**Figure 3.** Confusion matrix and classification report.

**Table 3.** Confusion matrix and classification report for the training dataset.

| Attributes | Precision | Recall | F1-Score | Support |
|:---:|:---:|:---:|:---:|:---:|
| 1 | 0.83 | 0.83 | 0.82 | 30 |
| 2 | 077 | 0.83 | 0.80 | 36 |
| 3 | 0.88 | 0.76 | 0.82 | 38 |
| 4 | 0.69 | 0.89 | 0.78 | 37 |
| 5 | 0.74 | 0.65 | 0.69 | 43 |
| 6 | 0.70 | 0.74 | 0.72 | 38 |
| 7 | 0.76 | 0.70 | 0.73 | 40 |
| 8 | 0.82 | 0.75 | 0.78 | 36 |
| 9 | 1.00 | 0.83 | 0.91 | 35 |
| 10 | 0.86 | 1.00 | 0.92 | 30 |
| 11 | 0.86 | 0.86 | 0.86 | 36 |
| 12 | 0.94 | 0.86 | 0.90 | 35 |
| 13 | 0.79 | 0.94 | 0.86 | 36 |
| 14 | 1.00 | 0.75 | 0.86 | 36 |
| Accuracy | | | 0.82 | 506 |
| Macro avg | 0.83 | 0.83 | 0.82 | 506 |
| Weighted avg | 0.83 | 0.82 | 0.82 | 506 |

Table 3 illustrates the deep learning performance metrics results, including the number of occurrences of each particular class in the true responses. As we can see in the support column of Table 3, the model was able to correctly identify actual incidences of the 14 elicitation techniques in a good ratio. Next, the results revealed the model's ability to return more relevant results than irrelevant ones, which is reflected by the macro-average precision score of 0.83 computed without considering the proportion and the weighted-average precision score of 0.83 computed by taking the mean of all per-class support relative to the sum of all support values. Next, the model returned the most relevant results, reflected by the macro-average recall score of 0.83 computed without considering the proportion, and a weighted-average recall score of 0.82 compute by taking the mean of all per-class support relative to the sum of all support values. Moreover, the model revealed a relatively higher macro-average F1 score of 0.82 computed without considering the proportion, and a weighted-average F1 score of 0.82 compute by taking the mean of all per-class support relative to the sum of all support values. Finally, calculating the accuracy of the prediction that was made correctly by the model. The results revealed a relatively higher accuracy ratio of 0.82 representing the model's prediction ability and efficiency in nominating a suitable elicitation technique based on the elicitation case scenario.

Model Validation

This section summarizes the evaluation metrics used to validate the model prediction. The deep learning model was validated using a loss curve and the area under the ROC curve (AUC).

Loss Curve

In this model, we implement the Mean Squared Error (MSE) as follows in Equation (9) to measure the amount of error in the deep learning model by calculating the average squared difference between the actual and predicted data point values.

$$\text{MSE} = \frac{1}{n} \sum_{i=1}^{n} \left( Y_i - \hat{Y}_i \right)^2 \tag{9}$$

where $Y_i$ is actual data point values, $\hat{Y}_i$ is the predicted data point values and $n$ is the total number of data point in the dataset.

Figure 4 shows the loss curve of the deep learning model for over 80 iterations. In contrast, the cost value decreases with every iteration during the neural network training session. Therefore, reflecting the learning performance over time in terms of experience. Finally, the cost value reached fewer than 0.3 points, which is considered an acceptable MSE score.

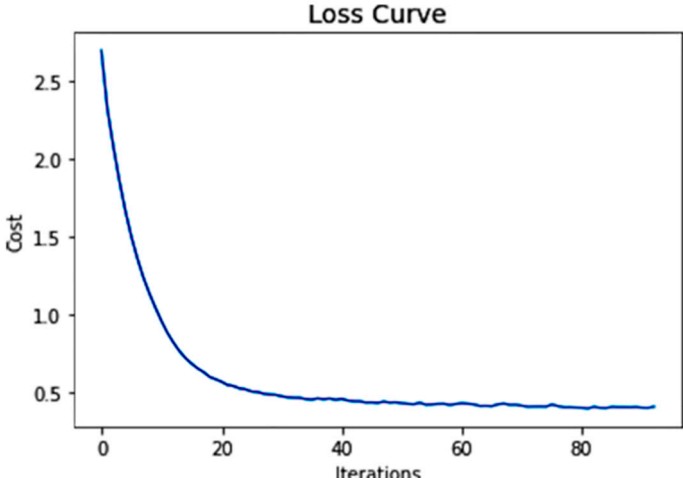

**Figure 4.** Loss curve.

The Area under the ROC Curve (AUC)

The area under the ROC curve (AUC) was used to calculate the deep learning model prediction accuracy. Thus, we computed the k-Fold cross-validation to maximize the use of the available dataset for training and assessing model performance. The key configuration parameter for k-fold cross-validation is k = 10, which defines the number folds in which to split a given dataset. The reason for this is that studies were performed and k = 10 was found to provide good trade-off of low computational cost and low bias in an estimation of model performance. Thus, our model scored 0.82 ($\pm$0.10), reflecting less statistical noise and hence a more reliable model performance.

Table 4 shows the One-vs-Rest and One-vs-One for multi-class classification scores of the deep learning model. In order to provide an aggregate measure of performance across all possible classification thresholds, One-vs-Rest was computed by comparing each class against all of the others at the same time. Next, One-vs-One was computed by comparing all possible two-class combinations of the dataset, thereby splitting the multi-class classification dataset into binary classification. The results illustrated in Table 4 confirm the model's ability to distinguish between the elicitation techniques with a score of 0.75.

**Table 4.** ROC AUC scores.

| One-vs-One ROC AUC Scores | One-vs-Rest ROC AUC Scores |
| --- | --- |
| 0.745365 (macro) | 0.754880 (macro) |
| 0.750311 (weighted by prevalence) | 0.755098 (weighted by prevalence) |

Figure 5 shows the areas under the multiclass ROC curves (AUC), evaluating the model's classification ability. Each colored line of the figure represents a specific elicitation technique in the model. The distribution of elicitation technique values between a false positive rate of zero and a true positive rate of one evidently illustrates the model's capacity to clearly distinguish between these techniques, confirming the model's ability to provide an accurate range of predictions.

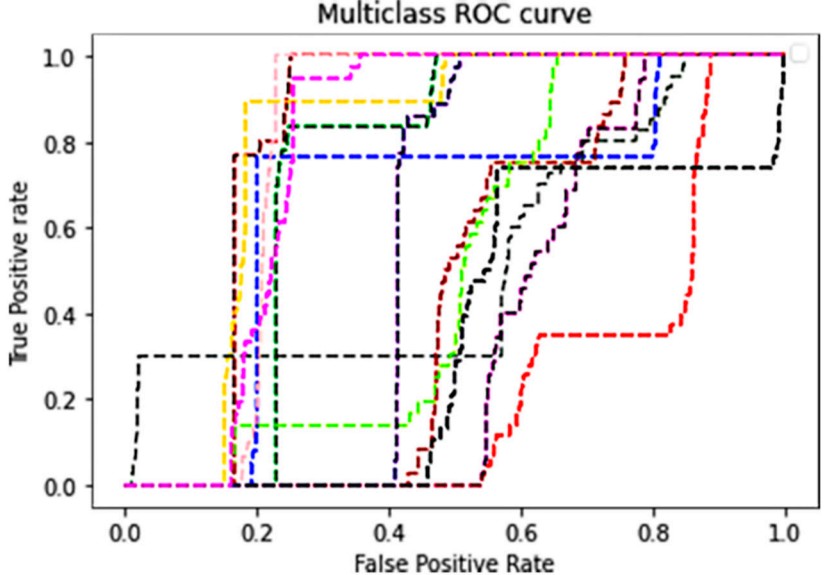

**Figure 5.** Multiclass ROC curve.

## 6. Discussion

This study aimed to utilize the neural network's deep learning ability in developing a robust decision-making mechanism for selecting suitable requirement elicitation techniques based on elicitation case scenario parameters, further automating the process to reduce human intervention in the selection process. This succeeds in avoiding human error occurrences due to a lack of experience. The deep learning model consisting of 3 hidden layers (each containing 100 neurons) was able to train the dataset containing the technique selection attributes samples for the 14 elicitation techniques. The neural network training was successful in depicting a higher learning curve in contrast to the loss curve. On the other hand, Figure 4 shows that the error rate decreases with every training iteration. Hence, the deep learning model's ability to predict suitable elicitation techniques has improved, reaching an accuracy ratio of 82%. Additionally, we sought to evaluate the results using the confusion matrix and the area under the ROC curve (AUC). As such, Table 3 of the confusion matrix illustrates the deep learning model's prediction ability to return more relevant results than irrelevant ones, something which can be attributed to its precision score of 0.83. It also reveals the model's ability to return most of the relevant results, which is attributed to its recall score of 0.83. Moreover, its cross-validation score is 0.82 ($\pm$ 0.10), indicating the dataset's integrity for training and assessing model performance. Moreover, the area under the ROC curve (AUC) was implemented to analyze the deep learning model prediction accuracy. Table 4 reveals the deep learning model's One-vs-One ROC AUC score of 0.74 and One-vs-Rest ROC AUC score of 0.75 for each label. Therefore, our results are a confirmation of the model's ability to distinguish between the elicitation techniques.

Furthermore, Figure 5 clearly illustrates the distributions pattern of the values falling in between the false positive rate of zero and true positive rate of one, once again confirming the previous results. Finally, these results appear to be consistent with each other. The deep learning model addresses the limitations raised in the literature review section, providing a robust decision-making model which can take advantage of the enhanced dataset. This further reduces the consequences of human error. The proposed deep learning model has the potential to improve the requirement elicitation process and increase the IS project's success rates, thus lowering risks by introducing a transferable solution through neural networks (such as an automated systematic decision-making model that will assist every requirement engineer in the field to select the most suitable elicitation techniques for the given scenario).

## 7. Conclusions

The primary aim of this study was to develop an intelligent, robust decision-making model for suitable requirement elicitation technique selection. Our model uses deep learning technology to automate the elicitation technique selection operation and reduce current human intervention errors. This improves software and IS project practices and reduces the risk of failure. The deep learning model in Figure 1 begins by categorizing the technique selection attributes (Tables 1 and 2) in an effort to identify the key decision-making factors. Next, we performed a comprehensive data collection process from various data sources to maximize the existing datasets. Moreover, scaling and mapping the collected dataset as part of the preprocessing phase was carried out to ensure the integrity of the data. Next, the neural network training session phase (Figure 2) was initialized with a configuration of 3 hidden layers, each containing 100 neurons. As a result, the model produced a prediction accuracy of 82%. Moreover, it had a precision and recall score of 0.83 and F1 score of 0.82. These findings show the model's ability to return more relevant results than irrelevant ones and return most of the relevant results. The model validation phase computed the loss curve and the area under the ROC curve (AUC), evaluation metrics which are used to validate deep learning model prediction. In Figure 3, the loss curve highlighted the decreasing cost value of every iteration, indicating increased performance over time for the model. Moreover, 10-fold cross-validation was used. Each fold is used as a testing set in the evaluation process. Furthermore, the area under the ROC curve (AUC) was computed to evaluate the model prediction accuracy. Table 4 illustrates One-vs-One ROC AUC scores of 0.74 and One-vs-Rest ROC AUC scores of 0.75 for each label. Moreover, Figure 5 showing the multiclass ROC curves (AUC) highlighted the model's ability to distinguish between the elicitation technique. These results confirm the model's ability to produce accurate predictions. However, two limitations still exist in this study. The first is that the model prediction is based on elicitation case scenario parameter rather than project parameters. The second limitation is the scale of the model's deployment. Overall, the proposed model is able to select a suitable elicitation technique that best fits the elicitation scenario, thus lowering the risks of project failure and improving the elicitation industry practice. Future studies should aim to increase the data collection sample size to accommodate further IS project environments. Furthermore, investigations should be conducted to broaden the model suitability feature to include IS project as an elicitation entity. Moreover, in the future, representation learning and label learning [28,29] should be examined to automate the elicitation technique selections operation. The implications of this study could be used to promote the automatization of the requirement elicitation process, thus increasing the potential for enhancing the produced systems designs.

**Author Contributions:** Conceptualization, H.D. and M.A. (Mohammed Abaker); methodology, H.D.; software, H.D.; validation, H.D., M.A. (Mohammed Abaker); formal analysis, H.D.; resources, H.D., M.A. (Mohammed Abaker), A.A. (Abdelzahir Abdelmaboud), M.A. (Mohammed Alghobiri); data curation, H.D., M.A. (Mohammed Abaker); writing—original draft preparation resources, H.D., M.A. (Mohammed Abaker); writing—review and editing, H.D., M.A. (Mohammed Abaker), A.H., H.E., N.A., M.A. (Mohammed Alghobiri), A.A. (Ahmed Abdelmotlab), A.A. (Abdelzahir Abdelmaboud);

supervision, H.D., M.A. (Mohammed Abaker); project administration, H.D., M.A. (Mohammed Abaker); funding acquisition, H.D., M.A. (Mohammed Abaker). All authors have read and agreed to the published version of the manuscript.

**Funding:** This research was funded by the Deanship of Scientific Research at King Khalid University, grant number (RGP.1/250/43).

**Institutional Review Board Statement:** Not applicable.

**Informed Consent Statement:** Not applicable.

**Data Availability Statement:** Not applicable.

**Acknowledgments:** The authors extend their appreciation to the Deanship of Scientific Research at King Khalid University for funding this work through General Small Groups Research Project under grant number (RGP.1/250/43).

**Conflicts of Interest:** The authors declare no conflict of interest.

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
