# Peer review of "Deep Learning Model for Selecting Suitable Requirements Elicitation Techniques"

_applsci, doi:10.3390/app12189060_

Round 1
Reviewer 1 Report
** Title **
- The initial letter of each notional word should be in uppercase.
Capitalization in Titles:
According to most style guides, nouns, pronouns, verbs, adjectives, and adverbs are the only words capitalized in titles of books, articles, and songs. Prepositions, articles, and conjunctions aren't capitalized (unless they're the first or last word).
â–ª In the title, all nouns, pronouns, adjectives, verbs, adverbs, and subordinating conjunctions (ex: if, because, that, which) should be capitalized.
â–ª Capitalize abbreviations that are otherwise lowercase (i.e., use DC, not dc or Dc) except for unit abbreviations and acronyms.
â–ª Articles (ex: a, an, the), coordinating conjunctions (ex: and, but, for, or, nor), and most short prepositions are lowercase unless they are the first or last word.
â–ª Prepositions of more than 3 letters (ex: before, from, through, with, versus, among, under, between, without) are capitalized.
** Abbreviations **
- ALL Abbreviations should be defined in full the first time they appear in the abstract, main text, and figure or table captions (Including title).
The abstract needs to be structured.
Literature section looks vague. Try to split paragraph in to several paragraphs.
Conclusion need to be effectively summarized.
Author Response
Dear Reviewer 1,
Thank you for giving me the opportunity to submit a revised draft of my manuscript titled Deep Learning Model for Selecting Suitable Requirements Elicitation Techniques to [Applied Sciences]. We appreciate the time and effort that you and the reviewers have dedicated to providing your valuable feedback on our manuscript. We are grateful to the reviewers for their insightful comments on our paper. We have been able to incorporate changes to reflect most of the suggestions provided by the reviewers. We have highlighted the changes within the manuscript.
Thank you for your consideration of this manuscript.
Sincerely,
Dr. Hatim Dafaalla
Assistant Professor, Department of Computer Science
King Khalid University

Reviewer 2 Report
Based on my opinion, the novelty of the paper is acceptable and can be used as a model of predicting elicitation technique. However, this manuscript requires consideration for following points:
1. In the introduction section, some challenges need to be included related to requirement elicitation phases so that neutral users will follow the contributions of your paper.
2. There are some typo mistakes in the paper. The authors are requested to thoroughly proofread the article.
3. Some references need to be revised.
4. Authors must restructure the abstract and mention the results achieved in the paper with clarity.
5. Figures can be more clear and visible.
6. First two tables have same Captions.
Author Response
Dear Reviewer 2,
Thank you for giving me the opportunity to submit a revised draft of my manuscript titled Deep Learning Model for Selecting Suitable Requirements Elicitation Techniques to [Applied Sciences]. We appreciate the time and effort that you and the reviewers have dedicated to providing your valuable feedback on our manuscript. We are grateful to the reviewers for their insightful comments on our paper. We have been able to incorporate changes to reflect most of the suggestions provided by the reviewers. We have highlighted the changes within the manuscript.
Thank you for your consideration of this manuscript.
Sincerely,
Dr. Hatim Dafaalla
Assistant Professor, Department of Computer Science
King Khalid University

Reviewer 3 Report
This manuscript applsci-1890760 presented a deep learning model for selecting suitable requirements elicitation. An experiment of the study was conducted on a collected dataset of 1684 technique selection attributes samples for the 14 elicitation techniques. The study adopted three criteria to evaluate the predictive model performance, confusion matrix, the area under a receiver operating characteristic curve (AUC), and the loss curve. The model scored prediction accuracy of 82%, One-vs-One ROC AUC scores of 0.74, and One-vs-Rest ROC AUC scores of 0.75 for each label. This result indicates the model's high prediction ability. The model provides a robust decision-making process to deliver the correct elicitation technique and lower the risk of projects failure. The implication of this study used to promote automatization of the elicitation technique selection process. Thus, enhancing the current requirement elicitation industry practices. My overall impression of this paper is that it is in general well-organized. It was a pleasure reviewing this work and I can recommend it for publication in Applied Sciences after a major revision. I respectfully refer the authors to my comments below.
1. The English needs to be revised throughout. The authors should pay attention to the spelling and grammar throughout this work. I would only respectfully recommend that the authors perform this revision or seek the help of someone who can aid the authors. For example,
---(Line 106) The term “Similar to P. Vitharana et al. A. K. Gupta, and A. Deraman [11].” is not a sentence.
2. (Page 2, Section 1. Introduction, last paragraph) The reviewer suggest to add “main contributions” and list clearly by breaking it down into three points. The reader can understand your contribution easily.
3. (Page 2, Line 59) The original statement “… Without doubt, successful implementation of the elicitation technique selection process using machine learning …” is suggested to revised as “… Without doubt, successful implementation of the elicitation technique selection process using machine learning (1) mfdnet: collaborative poses perception and matrix fisher distribution for head pose estimation (2) arhpe: asymmetric relation-aware representation learning for head pose estimation in industrial human-machine interaction, …”.
4. All the figures are not very clear. Please redraw the figure as the high-resolution format.
5. (Page 5, Figures 1-2) The reviewer suggests authors introduce clearly the overall flow of Figures 1-2 (in the body or in the picture description).
6. Experimental pictures or tables should be described and the results should be analyzed in the picture description so that readers can clearly know the meaning without looking at the body. For example, describe the colorful curves in Figure 5 and describe the results of the analysis of this phenomenon.
7. (Page 3, Section 2 Related Study) The original statement is suggested as “Although machine learning and deep learning ((1) DOI: 10.1109/TNNLS.2021.3055147, (2) edmf: efficient deep matrix factorization with review feature learning for industrial recommender system, (3) anisotropic angle distribution learning for head pose estimation and attention understanding in human-computer interaction) were used to implement the mechanism….”
8. (In all the Tables) Add a new table to demonstrate the scores of all the comparing methods. And the best scores should be bolded.
9. (Page 1, Introduction) The reviewer suggests authors don't list a lot of related tasks directly. It is better to select some representative and related literature or models to introduce with certain logic. For example, the latter model is an improvement on one aspect of the former model.
10. (Page 11, Line 412) “… and improving the elicitation industry practice.” is suggested as “… and improving the elicitation industry practice. In the future, the representation learning and label learning <1-2> will be examined to automate the elicitation technique selections operation” (CARM: Confidence-aware recommender model via review representation learning and historical rating behavior in the online platforms;; ngdnet: nonuniform gaussian-label distribution learning for infrared head pose estimation and on-task behavior understanding in the classroom)
11. The authors are suggested to add some experiments with the methods proposed in other literatures, then compare these results with yours, rather than just comparing the methods proposed by yourself on different models.
My overall impression of this manuscript is that it is in general well-organized. The work seems interesting and the technical contributions are solid. I would like to check the revised manuscript again.
Author Response
Dear Reviewer 3,
Thank you for giving me the opportunity to submit a revised draft of my manuscript titled Deep Learning Model for Selecting Suitable Requirements Elicitation Techniques to [Applied Sciences]. We appreciate the time and effort that you and the reviewers have dedicated to providing your valuable feedback on our manuscript. We are grateful to the reviewers for their insightful comments on our paper. We have been able to incorporate changes to reflect most of the suggestions provided by the reviewers. We have highlighted the changes within the manuscript.
Thank you for your consideration of this manuscript.
Sincerely,
Dr. Hatim Dafaalla
Assistant Professor, Department of Computer Science
King Khalid University

Reviewer 4 Report
This manuscript introduces a deep learning model for selecting suitable requirements elicitation with collected dataset of 1684 technique selection attributes samples for the 14 elicitation techniques. This study is important for addressing the importance of IS and software development projects. The core idea seems interesting, but the paper should be improved in some regards:
What is the main motivation of this study? What will the gap be addressed here? Various researchers have studied the deep learning model. What is the framework proposed here? How is the integration of deep learning model with Artificial Neural Network? A more explanation of the proposed technique should be given. Please explain a few more works that are related to this technique. What is the significant difference between this manuscript with other techniques?
Abstract should be improved: Give full name for the abbreviated IS for the first time. Is it only adopted three evaluation criteria? Properly name ROC AUC.
Section 3: Lines 192 & 193: Here mentioned that model validation. Accuracy? Clearly state which one is evaluation/performance metric, etc. Figure 1 should be explained clearly. What is MLP, ReLU. Why is there Reliability & Error Rate? Please explain the material to obtain the dataset & the scope of the study.
Section 4: Figure 2 should be explained in detail. What is the output there? Section 4.1: What is the research framework? Should explain ReLU. Eq. (4): What is value ‘a’? Describe clearly. Section 4.2: Line: 280: Clearly describe the confusion matrix. Lines 292-295: Clearly show the performance metrics equation for Precision, Recall, etc. Table 3: What is Support? Explain accuracy, macro avg, weighted avg? What is the value for Precision & Recall (accuracy)? Line: 334: explain why k=4? Table 4: Explain ‘one-vs-one’ & ‘one-vs-rest’?
Section 5: Description of limitations and future studies should be in Section 6: Conclusions.
Sections 5 & 6 should be improved. Some are redundance and repeated the same explanation. It would be better to see more implication and justification of the findings for all the analysis & results rather than just reporting the results. Should highlight the conclusion/significant contribution and impact of this study.
Grammar/English/Typo Errors:
Lines 71 & 72: Section 4 analysis…? Section 5 discussion?
Line 187: Data preparation…?
Line 209: ‘.’ as shown...
Line 214: ‘,’ Through…
Line 228: Table 1 Technique parameters should be Table 2
Line 233: Figure 2. Introduces…
Lines 289 & 298: Caption for Figure 3 & Table 3 are similar?
Line 325: Should be Figure 4.
Line 331: AUC?
Author Response
Dear Reviewer 4,
Thank you for giving me the opportunity to submit a revised draft of my manuscript titled Deep Learning Model for Selecting Suitable Requirements Elicitation Techniques to [Applied Sciences]. We appreciate the time and effort that you and the reviewers have dedicated to providing your valuable feedback on our manuscript. We are grateful to the reviewers for their insightful comments on our paper. We have been able to incorporate changes to reflect most of the suggestions provided by the reviewers. We have highlighted the changes within the manuscript.
Thank you for your consideration of this manuscript.
Sincerely,
Dr. Hatim Dafaalla
Assistant Professor, Department of Computer Science
King Khalid University

Reviewer 5 Report
The authors of the article explore the use of neural networks in building a deep learning model for selecting suitable requirements elicitation techniques.
The introduction is written in a coherent and logical manner. The presentation of methodology strategies, technique selection attributes, data collection and analysis is coherent and logical, as well as correctly discussed. Only the numbering of the included tables and figures needs improvement.
The discussion is presented correctly and clearly. The conclusions are logical and consistent with the presented analytical results.
The number of sources used is sufficient.
Author Response
Dear Reviewer 5,
Thank you for giving me the opportunity to submit a revised draft of my manuscript titled Deep Learning Model for Selecting Suitable Requirements Elicitation Techniques to [Applied Sciences]. We appreciate the time and effort that you and the reviewers have dedicated to providing your valuable feedback on our manuscript. We are grateful to the reviewers for their insightful comments on our paper. We have been able to incorporate changes to reflect most of the suggestions provided by the reviewers. We have highlighted the changes within the manuscript.
Thank you for your consideration of this manuscript.
Sincerely,
Dr. Hatim Dafaalla
Assistant Professor, Department of Computer Science
King Khalid University

Round 2
Reviewer 1 Report
All comments are addressed properly.
Author Response
Dear Reviewer 1,
Thank you for giving me the opportunity to submit a revised draft of my manuscript titled Deep Learning Model for Selecting Suitable Requirements Elicitation Techniques to Applied Sciences. We appreciate the time and effort that you and the reviewers have dedicated to providing your valuable feedback on our manuscript. We are grateful to the reviewers for their insightful comments on our paper. We have been able to incorporate changes to reflect most of the suggestions provided by the reviewers. We have highlighted the changes within the manuscript.
Thank you for your consideration of this manuscript.
Sincerely,
Dr. Hatim Dafaalla
Assistant Professor, Department of Computer Science
King Khalid University

Reviewer 4 Report
Some of my previous (round 1) comments yet to address by the authors such as:
Point 5: Is it only adopted three evaluation criteria?
--> In abstract, authors mentioned about using three criteria to evaluate
but in the context (page 8), using more than three criteria?
Point 6: Lines 192 & 193: Accuracy? Clearly state which one is evaluation/performance metric, etc. Figure 1 should be explained clearly.
--> Authors should clearly state which performance metric are used in this manuscript. Figure 1 also yet to explain clearly in the context. What is Reliability in Validation Parameters?
Point 9: Section 4: Figure 2 should be explained in detail. What is the output there?
Table 3: What is Support? Explain accuracy, macro avg, weighted avg? Line: 334: explain why k=4? Table 4: Explain ‘one-vs-one’ & ‘one-vs-rest’?
--> In Table 3, what is 'Support' there? and the Accuracy, Macro avg, & Weighted avg. Please explain in the context. Please check the suitability of caption Table 3. Yet to address the comment on why choosing k=4. Also for Table 4, explain the difference between ‘one-vs-one’ & ‘one-vs-rest’.
Author Response
Dear Reviewer 4,
Thank you for giving me the opportunity to submit a revised draft of my manuscript titled Deep Learning Model for Selecting Suitable Requirements Elicitation Techniques to Applied Sciences. We appreciate the time and effort that you and the reviewers have dedicated to providing your valuable feedback on our manuscript. We are grateful to the reviewers for their insightful comments on our paper. We have been able to incorporate changes to reflect most of the suggestions provided by the reviewers. We have highlighted the changes within the manuscript.
Thank you for your consideration of this manuscript.
Sincerely,
Dr. Hatim Dafaalla
Assistant Professor, Department of Computer Science
King Khalid University
